# The Evaluation of Response to Immunotherapy in Metastatic Renal Cell Carcinoma: Open Challenges in the Clinical Practice

**DOI:** 10.3390/ijms20174263

**Published:** 2019-08-30

**Authors:** Alessandra Raimondi, Giovanni Randon, Pierangela Sepe, Melanie Claps, Elena Verzoni, Filippo de Braud, Giuseppe Procopio

**Affiliations:** Department of Medical Oncology, Fondazione IRCCS Istituto Nazionale dei Tumori, 20133 Milan, Italy

**Keywords:** renal cell carcinoma, immunotherapy, RECIST, irRECIST, metastasis

## Abstract

Immunotherapy has changed the therapeutic scenario of metastatic renal cell carcinoma (mRCC), however the evaluation of disease response to immune-checkpoint inhibitors is still an open challenge. Response evaluation criteria in solid tumors (RECIST) 1.1 criteria are the cornerstone of response assessment to anti-neoplastic treatments, but the use of anti-programmed death receptor 1 (PD1) and other immunotherapeutic agents has shown atypical patterns of response such as pseudoprogression. Therefore, immune-modified criteria have been developed in order to more accurately categorize the disease response, even though their use in the everyday clinical practice is still limited. In this review we summarize the available evidence on this topic, with particular focus on the application of immune-modified criteria in the setting of mRCC.

## 1. Introduction

Renal cell carcinoma (RCC) represents about 3% of solid tumors worldwide, with an estimated incidence of approximately 330,000 new cases per year. In around one third of cases, RCC is diagnosed at an advanced stage, and another 30% of patients ultimately develop metastases after initial nephrectomy [1,2]. The prognosis of metastatic RCC (mRCC) is generally considered poor, with a predicted survival rate at 5 years inferior to 20%, even though the median survival time ranges from 7.8 to 43.2 months, through a stratification according to several prognostic factors [3]. 

In the last decades, thanks to a deeper understanding of the molecular biology of RCC, the therapeutic decision-making approach to mRCC has dramatically changed after the introduction of novel drugs [4,5]. First, the anti-angiogenic therapies targeting the Vascular Endothelial Growth Factor and its receptor, specifically bevacizumab and sunitinib, pazopanib, cabozantinib, sorafenib, axitinib and tivozanib, respectively, entered into the therapeutic algorithm of mRCC, leading to a significant improvement in the survival outcomes and quality of life [6].

Subsequently, the advent of immunotherapy, specifically with the immune-checkpoint inhibitors (ICIs) targeting the programmed death receptor 1 (PD1) and its ligand (PD-L1), deeply impacted on the treatment management of several cancers, such as non-small cell lung cancer (NSCLC) and melanoma, as well as RCC, providing new therapeutic opportunities for the patients [7,8,9]. 

The first drug that proved its safety and activity in both clinical trials and real-world settings was the anti-PD1 nivolumab [8]. In the CheckMate 025 study, nivolumab conferred a significant and clinically relevant improvement in survival as compared to everolimus with a much more manageable safety profile [10,11], leading the Food and Drug Administration (FDA) and the European Medical Association to the approval of this drug for the treatment of mRCC patients who had received a prior line with anti-angiogenic agents [12,13]. 

In order to optimize the potential benefit of immunotherapy in mRCC, several clinical trials have been designed and conducted to investigate the combinations of immune agents (i.e., nivolumab plus ipilimumab) or ICIs with anti-vascular agents (i.e., avelumab plus axitinib, pembrolizumab plus axitinib and atezolizumab plus bevacizumab) in the first-line setting, and the first results recently published showed an overall improvement of the treatment outcomes over the anti-vascular tyrosine kinase inhibitor monotherapy [14,15,16,17]. In detail, in the CheckMate 214 trial [14], a significant benefit in overall survival (OS) and overall response rate (ORR) for the combination of the anti-PD1 nivolumab plus the anti- cytotoxic T-lymphocyte antigen 4 (CTLA4) agent ipilimumab over sunitinib was shown. Moreover, in the Javelin Renal 101 study [15] a 6.6-month increase in terms of progression free survival (PFS) was demonstrated for the anti-PD-L1 agent avelumab plus axitinib as compared to sunitinib monotherapy. The IMmotion151 [17] trial reported that the combination of the anti-PD-L1 atezolizumab plus bevacizumab prolonged PFS versus sunitinib, and the KEYNOTE-426 [16] study showed a significant benefit in terms of OS, PFS and ORR for the anti-PD1 agent pembrolizumab plus axitinib as compared to sunitinib. 

In light of these data, the therapeutic scenario of mRCC is rapidly changing, but, unfortunately, several unmet needs remain in this setting. First of all, since only a small proportion of patients obtain a significant and long-lasting benefit from immunotherapy, the identification of predictive biomarkers for response to immunotherapeutic agents, able to refine the patients’ selection, represents an open challenge [18,19]. Secondly, the optimal choice of the criteria to evaluate the tumor response to ICIs is still a matter of debate, since significant evidence highlighted that the response evaluation criteria in solid tumors (RECIST) 1.1 criteria might misinterpret the disease and underestimate the rate of patients deriving benefit from immunotherapy [20]. 

In this review we describe the available tools to categorize the tumor response to immunotherapeutic agents and we summarize the evidence collected on this topic with a specific focus on mRCC.

## 2. Molecular Mechanisms of Action of Immune-Checkpoint Inhibitors

Immune-checkpoint receptors are key factors in the immune system that act as negative regulators by limiting the proliferation and activation of immune cells, particularly T cells, but even macrophages and Natural Killer cells. The most studied immune-checkpoints involved in the tumor biological scenario are the PD1/PD-L1 and the CTLA4 pathways, that have been targeted by a number of drugs developed and investigated in clinical trials in several tumor settings and recently entered into the clinical practice for specific neoplasms [21]. 

The CTLA4 and PD1/PD-L1 axis are fundamental in the complex scenario of the finely tuned regulation of the immune system. In detail, they play a key role in the modulation of adaptive immunity and are endowed with complementary functions. First of all, during the T-cells activation phase, the CTLA4 antigen is upregulated on the plasmatic membrane of T-cells and it binds with high affinity and avidity to the ligands B7 (the same target of the co-stimulatory receptor CD28) expressed on the surface of the professional Antigen Presenting Cells (APCs), resulting in an inhibition of T-cell activation through a downstream signaling pathway [22]. While the CTLA4 checkpoint is involved in the early activation of T cells, the PD1/PD-L1 one is located in a subsequent step of the immune response. Specifically, PD1 is a transmembrane protein selectively expressed upon the activated effector T cells and, thanks to the binding with its two ligands (PD-L1 and PD-L2), expressed on various cells including tumor, APCs and T cells, it inhibits the signaling pathways leading to an effective T-cell response and limits the T cells activity during the inflammatory response [23].

Given the crucial role of T cells in the anti-cancer immune defense, increasing evidence has been collected that cancer is endowed with the capacity to induce an immune-suppressive reaction leading to the creation of a *milieu* favorable for tumor growth and progression. Since tumors may be able to utilize the immune-checkpoint pathways to escape from the T-cell based anti-cancer immunity, antibodies specifically targeting these key mediators of immune response, the “ICIs”, have been developed on the basis of this biologic rationale [24]. In detail, they determine an action of “releasing the brakes” of the immune system, where the anti-CTLA4 agents are able to contrast the inactivation of the immune response and to stimulate the induction of an anti-neoplastic immune reaction, while the anti-PD1 and anti-PD-L1 drugs act by enhancing the effector activity of T cells in the peripheral tissues, most importantly in the tumor microenvironment, to selectively recognize and kill cancer cells [7,9].

The anti-CTLA4 antibody ipilimumab was the first ICI that entered the clinical trials in cancer patients and it provided practice-changing results in advanced/metastatic melanoma patients, since it showed remarkable efficacy data in phase I, II and III trials, inducing durable responses both in the first and further line setting, leading to the FDA approval in 2011 [25]. While in advanced melanoma, ipilimumab revolutionized the therapeutic management of patients, in mRCC it did not demonstrate a clinically meaningful benefit at the price of a relevant toxicity burden. In detail, in a phase II trial, a total number of 61 patients received a high or low dose of ipilimumab for up to a year of treatment, obtaining a 12.5% and 5% response rate, respectively, and no evidence of complete responses or long-lasting disease regressions, with a 43% and 18% grade 3, 4 or 5 drug-related adverse events rate, respectively [26].

For what concerns the PD1/PD-L1 checkpoint, the anti-PD1 agents, among which nivolumab and pembrolizumab, and the anti-PD-L1 drugs, such as atezolizumab, avelumab and durvalumab, paved the way for a radical change in the therapeutic algorithm of several tumors, like melanoma, NSCLC and mRCC, as well as urothelial carcinoma, Merkel Cell Carcinoma and Hodgkin lymphoma. Specifically, in advanced/metastatic melanoma, the anti-PD1 agents proved to confer a higher efficacy with a more favorable safety profile as compared to ipilimumab, therefore both nivolumab and pembrolizumab are currently approved and widely used drugs in the clinical practice [27]. Moreover, in NSCLC, ICIs targeting PD1/PD-L1 proved a significant OS benefit over the conventional chemotherapy, and, since 2015, nivolumab, pembrolizumab and atezolizumab were approved by FDA after the first-line treatment, independently by the hystologic type, while pembrolizumab received the approval as front-line therapy in patients with >50% PD-L1 expression [28]. In the setting of mRCC, nivolumab entered the therapeutic scenario in previously treated patients in light of the results of the CheckMate025 trial. In detail, a significantly longer median OS was observed (median OS 25 versus 19.6 months with nivolumab versus everolimus) and a higher objective response rate (25% versus 5% in nivolumab and everolimus arm) was evidenced, while median PFS was 4.6 versus 4.4 months in nivolumab versus everolimus arms, respectively [11], leading to the design and conduction of new trials exploring immunotherapy and anti-angiogenic agents combinations in the first-line setting.

Finally, besides the anti-CTLA4 and anti-PD1/PD-L1 agents, several other immune-checkpoints are under investigations in order to potentiate the action of reshaping the immune system activity in the tumor setting and to optimize the outcomes of immunotherapy in cancer patients, with drugs acting both as blockers of the inhibitory regulators of the immune system, or as stimulators of the activating pathways. In detail, potential therapeutic targets are the lymphocyte activation gene 3 protein (LAG-3), killer-cell immunoglobulin- like receptor (KIR), T cell immunoreceptor with Ig and ITIM domains (TIGIT), and T cell immunoglobulin and mucin domain-containing 3 (TIM-3 or HAVCR2), the tumor necrosis factor receptor superfamily member 4 (TNFRSF4 or OX40 or CD134), tumor necrosis factor receptor superfamily member 18 (TNFRSF18), CD27, and CD137 [21].

Initial evidence has been collected, but, although the results appear to be promising, further research is needed in this setting to translate these data into the clinical practice.

## 3. Criteria for the Evaluation of Tumor Response to Cancer Treatments and New Challenges

The tumor response to cancer treatments is assessed and classified according to precise and standardized radiological criteria, used as surrogate for patient outcomes to guide physicians in the clinical decision-making [29]. In detail, the World Health Organization (WHO) group developed in 1981 the first widely accepted criteria, based on the mechanism of action of chemotherapy: a direct cytotoxic effect inducing cancer cell death and a consequent tumor shrinkage in case of response (partial and complete response, PR and CR), or a tumor growth and/or appearance of new lesions for refractoriness to treatment leading to disease progression (PD). These criteria are bidimensional and based on the concept of the assessment of tumor burden, by summing the products of the two largest perpendicular diameters (SPD) of all index lesions (five lesions per organ, up to 10 visceral lesions), and the determination of response to treatment by evaluating the changes from baseline [30]. Subsequently, the RECIST group provided novel criteria, published in 2000 (RECIST 1.0) and revised in 2009 (RECIST 1.1), that switched to a unidimensional measurement method (longest diameter of the non-nodal lesions and short axis for lymph nodes) of the tumor burden in order to improve the reproducibility and reduce the variability and potential misclassification of tumor response [31,32]. In the last decade, the majority of clinical trials conducted in cancer patients have based their endpoints upon the RECIST 1.1 criteria, however, the progress in tumor research led to the introduction of new classes of anti-neoplastic agents, such as targeted therapies and immunotherapy. Evidence has been collected highlighting the limitations of RECIST 1.1 criteria in accurately categorizing the tumor response to these new drugs, since their anti-tumor effects rely on different mechanisms of action, uncovering a novel major challenge for clinicians [29,33]. Focusing on immunotherapy, ICIs act by stimulating the immune system to build and deliver an anti-tumor immune response, therefore disease response may occur with atypical patterns and the traditional criteria could lead to a misinterpretation and underestimation of the real number of patients achieving benefit from immune-directed agents [24]. This highlights a relevant unmet clinical need and raises questions about how clearly determining which patients truly derive a benefit from immunotherapy and for whom the premature treatment discontinuation would be detrimental [34]. In detail, besides the conventionally defined tumor response, consisting of tumor shrinkage, the radiological response pattern in patients treated with ICIs could present as an early deep or even complete response, a prolonged disease stabilization before an ultimate tumor shrinkage, an initial increase in the tumor burden or a mixed response with appearance of new lesions followed by a delayed tumor response, and an early and rapid disease progression [35]. The last two conditions, named “pseudoprogression” and “hyperprogression”, respectively, are the most challenging scenarios, and have been described in several reports, although a precise and standardized definition has not been established yet [36,37]. Hyperprogression is a rapid increase in tumor growth rate after the start of immunotherapy, with important clinical implications for patients, and its pathogenesis and incidence are still to be fully characterized [38]. Pseudoprogression may find its underlying biological rationale in the mechanism of action of ICIs, and the first potential explanation is that the delayed onset of immune response could enable the tumor to initially grow before the activation of the anti-cancer immunity [39]. The second hypothesis is that, since these agents induce an immune response, the tumor lesions appear increased in size and previously radiologically undetectable tumor deposits are identified since they are “inflamed” for the infiltration of lymphocytes, while subsequently, after the resolution of inflammation and edema, tumor shrinkage is appreciated. This theory has been corroborated by translational studies including serial tumor biopsies in the disease course during the treatment with immunotherapy [40]. The incidence of pseudoprogression has been described in approximately 10% of patients with metastatic melanoma treated with anti-CTLA4 and anti-PD1 antibodies [41,42,43], whereas the frequency in other tumor settings is unclear, even though, in mRCC, recent data derived from patients treated beyond progression in the clinical trials report a comparable result, with a rate of about 5–15% [44,45].

## 4. Immune-Modified Criteria

In order to overcome the limitations of WHO and RECIST criteria to accurately assess and categorize the tumor response to immunotherapy, novel criteria have been developed by modifying the traditional ones on the basis of the specific mechanism of action of ICIs and the unique atypical patterns of response that they may induce [18]. Their definition and application in the clinical setting are described in the next session and summarized in Table 1.

### 4.1. Immune-Related Response Criteria (irRC)

The first immune-modified criteria to be developed were the irRC in 2009, that represented an evolution of the WHO criteria. They were evaluated through an analysis of the dataset of 3 phase II multicenter clinical trials investigating ipilimumab in patients with advanced melanoma, showing that 9.7% of patients, initially deemed to have PD with WHO criteria, ultimately achieved a response to ipilimumab [43]. The most remarkable changes consisted in the inclusion of new lesions in the tumor burden assessment and the confirmation at 4 weeks needed for tumor progression. In detail, the SPD of all index lesions is calculated at baseline, as in WHO criteria, and, at each subsequent assessment, the SPD of the index lesions and of new measurable lesions (≥ 5 × 5 mm; up to 5 new lesions per organ: 5 new cutaneous lesions and 10 visceral lesions) are combined to provide the total tumor burden. The tumor response is defined as irCR for complete disappearance of all lesions, irPR for decrease in tumor burden ≥50% versus baseline, irPD for increased in tumor burden ≥25% versus nadir, all requiring a confirmation, and irSD in the other cases. Additionally, the authors reported that patients with irSD showing slow-declining tumor burden ≥25% from baseline at the last tumor assessment should be considered as achieving a clinically meaningful response, although not meeting the criteria for irPR [43]. 

Afterwards, irRC were evaluated in the phase 1b Keynote-001 trial, investigating pembrolizumab in advanced melanoma, and the authors reported that RECIST 1.1 criteria underestimated the benefit of immunotherapy in a subgroup of patients, moreover the 2-year OS rate was 77.6% and 17.3% in patients with non-PD and PD per both criteria while 37.5% in those with RECIST 1.1 PD not confirmed by irRC [42]. However, the methodological differences between irRC and RECIST 1.1 could have partially jeopardized these results, since they are designed on bidimensional and unidimensional measurements, respectively, and with different thresholds [34].

### 4.2. Unidimensional irRC or Immune-Related RECIST (irRECIST) and Immune-Modified RECIST (imRECIST)

In 2013, the unidimensional irRC were presented as an evolution of the validated and widely used RECIST 1.1 criteria, incorporating the major changes introduced by the irRC criteria [46]. In detail, the new lesions evidenced in the disease re-evaluations were included in the sum of the measurements (≥10 mm longest diameters for non-nodal and ≥15 mm short axis for lymph nodes, ≤2 per organ and 5 total new lesions allowed at each time point). The response was defined as irCR for disappearance of all lesions, irPR for ≥30% decrease versus baseline and irPD for ≥20% increase versus nadir, requiring confirmation on two consecutive scans ≥4 weeks, ir-stable disease (irSD) in other cases [46,47]. In a casuistic of melanoma patients treated with ipilimumab, a high concordance between unidimensional irRC based on RECIST 1.0 and irRC was shown, with less measurement variability, due to the unidimensional method [46]. Afterwards, irRECIST were developed according to RECIST 1.1 criteria and the results were highly consistent [48]. The ultimately-defined unidimensional irRC criteria were retrospectively evaluated in a dataset of NSCLC patients treated with nivolumab, evidencing identical response rate and longer time to progression as compared with RECIST 1.1 criteria, and in advanced melanoma patients treated with pembrolizumab, showing a low rate of patients experiencing pseudoprogression and that a <20% increase in the tumor burden was associated with longer OS, possibly representing a marker of benefit from immunotherapy [49]. These criteria are usually referred to as irRECIST, and, despite the remarkable data presented, they have not been formally validated and consistently applied, thus possibly impairing the comparability of the results across the different studies [50].

Similarly, the recently presented imRECIST are based on the same concept of unidimensional irRC and their key principles are that the best overall response may occur after the radiologic PD in patients continuing treatment, that new lesions are included in the tumor burden when measurable, while not factored in the PD assessment when not measurable, and that progression in non-target lesions does not define PD [51]. ImRECIST were evaluated in a dataset of NSCLC, melanoma, RCC and urothelial cancer patients treated with atezolizumab, highlighting their potential to further refine the definition of the tumor response to immunotherapy [52].

### 4.3. Immune RECIST (iRECIST)

In 2017, the RECIST working group developed the iRECIST criteria with the aim of overcoming the heterogeneity of the previously defined criteria thanks to a rigorous and structured three-step protocol: publication of consensus guidelines, creation of a data warehouse hosting data from patients included in future clinical trials with immune-directed agents, and validation of the proposed criteria through the collected data [53]. In detail, the principles of RECIST 1.1 apply for the definition of target and non-target lesions (except for new lesions that should be subclassified in target and non-target) and the categorization of tumor response, however the major changes occur at the assessment of PD, that is renamed unconfirmed PD (iUPD). In this case, confirmation is required, consisting of either a further increase in size (or number of new lesions) in the lesion category in which PD was determined, or PD RECIST 1.1 in lesion categories not previously meeting PD criteria. In case that confirmed PD (iCPD) is not defined and tumor shrinkage occurs, meeting the criteria for iCR, iPR or iSD, the bar for the measurement is reset and the treatment may continue, provided that the patient is clinically stable [50]. 

The iRECIST criteria are waiting for a formal validation and the question about the concordance between the different immune-modified criteria is still unanswered. Recently, evidence has been collected upon the comparison of irRECIST and the newly-defined iRECIST in a retrospective dataset of NSCLC patients treated with anti-PD1 or anti-PD-L1 ICIs, showing a 3.8% discrepancy rate between the two criteria, accounting for discordances with a potential impact on the therapeutic decision-making [28]. 

## 5. Evidence in mRCC

The vast amount of the evidence upon this topic has been collected in melanoma and NSCLC. Regarding mRCC, the analysis of the subgroup of patients receiving the treatment beyond progression (TBP) in the two studies investigating nivolumab in the pretreated disease setting provided important data [54]. In detail, in the randomized phase II CheckMate 010 [8] trial of nivolumab at the dose of 0.3, 2 or 10 mg/kg every 2 weeks, TBP was allowed in case of good tolerability and the investigator’s judged clinical benefit from treatment. A global rate of 21% patients received TBP for more than 6 weeks and 7% of the intention to treat population had a ≥30% tumor shrinkage versus baseline after TBP [55]. Moreover, a subgroup of patients treated with nivolumab in the CheckMate 025 trial achieved a benefit from TBP, showing a tumor reduction after the first PD [56].

As for what concerns the potential application of the different immune-modified criteria in mRCC, unfortunately, very few data are available at the present time. First of all, the mRCC cohort of the PCD4989g trial (NCT01375842) [57] was included in the study that defined and evaluated the imRECIST criteria, showing consistent results with the other subgroups [52]. Recently, a secondary analysis of the CheckMate 010 trial [8] was published, aimed at investigating the irRECIST criteria for the evaluation of biomarkers of response to nivolumab in mRCC patients. The different endpoints assessed according to RECIST 1.1 and irRECIST criteria were compared and the results showed that immune-related PFS (irPFS) was significantly longer than median PFS per RECIST 1.1 (5.5 versus 3.3 months), unlike ir-ORR that did not significantly differ from ORR per RECIST 1.1 (22.8% versus 21%), even though PD according to RECIST 1.1 overestimated the irPD (35.3% versus 24.6%) [58].

These results are extremely promising, and in the ongoing and recently-presented clinical trials on immunotherapy and anti-vascular agents combinations in the first-line setting (Javelin Renal 101, Keynote 426 and IMmotion 151) [14,15,16,17] the irRECIST [46,48] are included in the secondary or exploratory endpoints. Therefore, in the near future, evidence will be collected on this topic, potentially providing the basis for the validation of immune-modified criteria in mRCC treated with immunotherapy.

## 6. Discussion

The acknowledgement of the limitations of the standardized and widely used RECIST 1.1 in defining the tumor response to the novel immunotherapeutic agents led to the development of immune-modified criteria, representing an evolution of the standard criteria based on the mechanisms of action of immunotherapy with ICIs and the atypical patterns of disease response that they induce [29]. However, given the heterogeneity of these immune-modified criteria and the lack of their extensive validation, the experts still recommend their application to the clinical trials rather than incorporating them into the real-world therapeutic decision-making, and to maintain the use of RECIST 1.1 criteria for the primary trials endpoint, leaving immune-modified criteria among the secondary or exploratory endpoints [50]. Nevertheless, the results from the secondary analysis of the ongoing clinical trials will provide useful data in this setting, potentially changing the scenario. 

Moreover, non-ICI immunotherapies, such as vaccines or adoptive cell transfer approaches, including chimeric antigen receptor (CAR)-T cells, lymphokine-activated killer cells, tumor-infiltrating lymphocytes, T-cell receptor transduced T cells, are currently under evaluation and they raise an additional challenge in this setting [59]. In fact, although the key action of these agents is focused on the immune system activation, their mechanisms of activity dramatically differ from ICIs, therefore we could expect potential brand-new pathways of response to treatment. Moreover, since the research progress led to the development of several therapeutic approaches with different possible biologic modifications, they could potentially produce a wide spectrum of radiologic scenarios, thus complicating the identification of patients achieving benefit from treatment even more. The future and ongoing research may provide clinically relevant data to shed a light on this topic.

Furthermore, few data are available upon the correlation between the radiological tumor response at the imaging and what happens in the neoplasia at the tissue-based level. In a recent proof-of-concept study of neoadjuvant nivolumab in early stage NSCLC, the RECIST response did not correlate with the pathological response, evidencing a major or complete pathological response in several cases of radiological SD [60]. Finally, the clinical evaluation of patients treated with immunotherapy is fundamental, mainly in case of unconfirmed PD, in order to identify which patients would benefit from treatment continuation and which are experiencing PD or even hyperprogression, although this phenomenon should be deeply understood and characterized [37]. This is peculiarly relevant in the everyday clinical practice, where the assessment of the patient’s conditions and of the clinical benefit achieved from treatment plays a key role in the therapeutic decision-making. In particular, an early discontinuation of treatment with ICIs in presence of an initial disease progression would not likely represent the optimal choice in case of symptomatic benefit and good or even improved clinical conditions. The clinical judgment of physicians is fundamental as long as the radiological assessment of disease response in order to design and tailor the optimal therapeutic algorithm in this setting.

In conclusion, a deeper characterization of the immune system and the complex pathway of the anti-tumor response induced by ICIs could help the development and consolidation of the immune-modified criteria, with the aim to accurately categorize the disease response and to optimize the treatment outcomes for cancer patients. 

## Figures and Tables

**Table 1 ijms-20-04263-t001:** Immune-modified criteria.

	RECIST 1.1	irRC	irRECIST	imRECIST	iRECIST
**Spatial Assessment**	Unidimensional	Bidimensional	Unidimensional	Unidimensional	Unidimensional
**Target lesions**	Sum of longest diameter of measurable lesions (i.e., ≥ 10 mm in diameter, 15 mm for lymph node lesions). Maximum 5 lesions (2 by organ site).	SPD of measurable lesions (≥ 5 × 5 mm^2^) of all index lesions. Maximum 5 lesions per organ, up to 10 visceral lesions and five cutaneous lesions.	SLD of measurable lesions (i.e., ≥ 10 mm in diameter).	SLD of measurable lesions (i.e., ≥ 10 mm in diameter, 15 mm for lymph node lesions). Maximum 5 lesions (2 by organ site).	SLD of measurable lesions (i.e., ≥ 10 mm in diameter, 15 mm for lymph node lesions). Maximum 5 lesions (2 by organ site).
**Non-target lesions**	Other than measurable disease. Contribute to CR and PD.	Only preclude irCR.	Only preclude irCR.	Only preclude imCR.	Other than measurable disease. Contribute to CR and PD.
**New lesions**	Always PD.	New measurable lesions are incorporated for calculating TB *.	The longest diameter of new measurable lesions is incorporated for calculating SLD.	New measurable lesions are incorporated for calculating SLD.	Result in iUPD. Characterized as measurable/non measurable according to RECIST 1.1. Not included in SLD.
**PD**	≥ 20% increase in the SLD from best response (at least ≥ 5 mm). Unequivocal progression of non-target lesions. Appearance of new lesions.	≥ 25% increase in TB compared with nadir.	≥ 20% increase in SLD compared with nadir.	≥ 20% increase in SLD compared with nadir.	iUPD is defined by first PD according to RECIST 1.1. iCPD if next assessment after iUPD reveals new lesions, increase size of new lesions (≥ 5 mm for target lesions and any increase in non target lesions).
**Confirmation of progression**	Not required.	Yes, by a repeated assessment at least 4 weeks apart.	Yes, in two consecutive observations, at least 4 weeks apart.	Yes, by a repeated assessment at least 4 weeks apart.	Yes, iUPD should be confirmed in a subsequent assessment, 4–8 weeks apart.
**CR**	Disappearance of all lesions, lymph nodes < 10 mm.	Complete disappearance of all lesions. Needs confirmation by a repeated assessment at least 4 weeks apart.	Disappearance of all lesions. Needs confirmation in two consecutive observations, at least 4 weeks apart.	Disappearance of all lesions.	Disappearance of all lesions. Can follow iUPD.
**PR**	≥30% decrease in the SLD from baseline.	≥50% decrease in TB compared to baseline. Needs confirmation by a repeated assessment at least 4 weeks apart.	≥30% decrease in SLD from baseline. Needs confirmation in two consecutive observations, at least 4 weeks apart.	≥30% decrease in SLD from baseline.	≥30% decrease in SLD from baseline. Can follow iUPD.
**SD**	Not meeting criteria for PD/PR/CR.	Not meeting criteria for PD/PR/CR.	Not meeting criteria for PD/PR/CR.	Not meeting criteria for PD/PR/CR.	Not meeting criteria for PD/PR/CR. Can follow iUPD.

**Abbreviations.** RECIST: response evaluation criteria in solid tumor. irRC: immune-related response criteria. irRECIST: immune-related RECIST. imRECIST: immune-modified RECIST. iRECIST: immune RECIST. CR: complete response. iCPD: immune-confirmed progressive disease. iUPD: immune-unconfirmed progressive disease. PD: progressive disease. PR: partial response. SD: stable disease. SLD: sum of the longest diameters. SPD: sum of the products of the two largest perpendicular diameters. TB: tumor burden. * TB is defined as TB = SPD _index lesions_ + SPD _new, measurable lesions_.

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
