# Peer review of "The Evaluation of Response to Immunotherapy in Metastatic Renal Cell Carcinoma: Open Challenges in the Clinical Practice"

_ijms, 2019, doi:10.3390/ijms20174263_

Round 1

Reviewer 1 Report

The review is comprehensive and informative for understanding growing field of immunotherapy in metastatic renal cell carcinoma. The review allowed me to understand upcoming PD-1 immune therapies for treating cancers with focus on RCC. The will help many peers in PD-1 and PDL-1 based therapeutics. Apart from rechecking the reference as per journal format there is no improvement needed.

Author Response

We deeply thank the Reviewer for his/her comments upon our manuscript.

We reformatted the references according to the Journal guidelines as requested.

Reviewer 2 Report

In the current review, Raimondi et al, did an excellent job in summarizing the current clinical use of immune-checkpoint inhibitors in metastatic renal cell carcinoma (mRCC), and discussed some of the mechanisms of action of the inhibitors. More importantly, authors discussed intensively on the RECIST and iRECIST criteria based on clinical data that will provide important inside in assisting therapeutic decision making.

Author Response

We deeply thank the Reviewer for his/her comments upon our manuscript.

Reviewer 3 Report

This excellent review gives a good overview on the problem of evaluating the response of immunotherapeutics, especially in metastasised renal cell carcinoma (mRCC). This is a topic which is highly up to date and which poses many difficulties to radiologists and oncologists. Due to the clinical relevance, this review focuses on immune checkpoint inhibitors, and especially anti-PD1, anti-PD-L1 and anti-CTLA4 antibodies. It gives a good overview on the topic and future developments, which can surely be expected. The linguistic style will be sufficient for publication after some minor adjustments. This work will find the interest of many readers. Therefore I strongly support publication.

I have however following suggestions for improving the manuscript:

The radiological response criteria play an central role in clinical studies. A multitude of experimental studies with immunotherapies different from checkpoint inhibitors (i.e. CAR-T cells, virotherapy etc) will be done in the next years. Please discuss more in detail the requirements and difficulties these studies might pose for the evaluation criteria. The citations 12 and 13 are not formatted correctly. Please change.

Author Response

We deeply thank the Reviewer for his/her comments upon our manuscript.

We agree with the Reviewer with the need to discuss the potential challenges raised by the new studies investigating non-checkpoint inhibitors immunotherapies and we added a paragraph to the discussion on this topic, as follows: “Moreover, non-ICI immunotherapies, such as vaccines or adoptive cell transfer approaches, including chimeric antigen receptor (CAR)-T cells, lymphokine-activated killer cells, tumor-infiltrating lymphocytes, T-cell receptor transduced T cells, are currently under study and they raise an additional challenge in this setting (Lin, Y.; Okada, H. Cellular immunotherapy for malignant gliomas. Expert Opin Biol Ther. 2016, 16, 1265-1275). In fact, although the key action of these agents is focused on the immune system activation, their mechanisms of activity dramatically differ from ICIs, therefore we could expect potential brand new pathways of response to treatment. Moreover, since the research progress led to the development of several therapeutic approaches with different possible biologic modifications, they could potentially produce a wide spectrum of radiologic scenarios, complicating even more the identification of patients achieving benefit from treatment. The future and ongoing research may provide clinically-relevant data to shed a light on this topic”.

Moreover, we reformatted the citations 12 and 13 according to the Journal guidelines.

Additionally, we revised the manuscript linguistic style as suggested.